# ENSEMBLER: COMBATING MODEL INVERSION ATTACKS USING MODEL ENSEMBLE DURING COLLABORATIVE INFERENCE

## ABSTRACT

Deep learning models have exhibited remarkable performance across various domains. Nevertheless, the burgeoning model sizes compel edge devices to offload a significant portion of the inference process to the cloud. While this practice offers numerous advantages, it also raises critical concerns regarding user data privacy. In scenarios where the cloud server's trustworthiness is in question, the need for a practical and adaptable method to safeguard data privacy becomes imperative. In this paper, we introduce *Ensembler*, an extensible framework designed to substantially increase the difficulty of conducting model inversion attacks for adversarial parties. *Ensembler* leverages model ensembling on the adversarial server, running in parallel with existing approaches that introduce perturbations to sensitive data during colloborative inference. Our experiments demonstrate that when combined with even basic Gaussian noise, *Ensembler* can effectively shield images from reconstruction attacks, achieving recognition levels that fall below human performance in some strict settings, significantly outperforming baseline methods lacking the *Ensembler* framework.

## 1 INTRODUCTION

In numerous critical domains, deep learning (DL) models have demonstrated exceptional performance when compared to traditional methods, including image classification Deng et al. (2009); Dosovitskiy et al. (2021), natural language processing Brown et al. (2020), protein predictions Jumper et al. (2021), and more. One noteworthy trend accompanying these impressive advances is the escalating size of DL models employed for these tasks (Hu et al., 2021), with the famous GPT-3 model containing 175 billion parameters (Brown et al., 2020). As a result, when tasks necessitate the involvement of edge devices such as mobile phones, reducing the computational workload on these devices becomes imperative. A prevalent approach involves offloading a substantial portion of the workload to a cloud server capable of executing extensive computations. This framework can be conceptualized as collaborative computing, where a client collaborates with a server offering computation-as-a-service (CaaS).

Recently, some attention in the research community has been shifted to an emphasis on the privacy of client's sensitive data in such a framework. While the client inherently trusts itself, the server may pose as an adversarial entity seeking to compromise the user's privacy during the inference process. This risk becomes particularly pronounced when DL models are tasked with handling sensitive data, such as disease classification or facial authentication, which require access to medical or facial user information. In other scenarios, a client could be a small company that holds private models and uses the server solely for the purpose of providing service. It also does not want the server to access the data of its customers, which sometimes contains sensitive information. With the prevalence of edge computing, there is an increasing need for researchers to develop a machine learning framework that supports secure, accurate, and efficient machine learning service, and works in this area are often categorized under the term privacy-preserving machine learning (PPML).

There have been multiple works addressing this formidable challenge of safeguarding the client's sensitive information in collaborative inference scenarios, an important part of the entire PPML framework. For an extensive discussion on different algorithmic and architectural choices and their

impacts on privacy protection, we refer readers to Section 5 and Table 2 of the comprehensive survey by (Xu et al., 2021). In this paper, we will simply group existing approaches into two categories: encryption-based algorithms that guarantee privacy at the cost of thousands of times of time efficiency (Mishra et al., 2020; Knott et al., 2021; Tan et al., 2021; Reagen et al., 2021; Rathee et al., 2020; Lam et al., 2023; Watson et al., 2022), and perturbation-based algorithms that operate on the intermediate layers of a DL architecture, introducing noise to thwart the adversary's ability to recover client input (Mireshghallah et al., 2020; Osia et al., 2018; Lu et al., 2022; Sirichotedumrong & Kiya, 2021). Since perturbation-based algorithms directly operate on the intermediate outputs from the client, they incur minimal additional complexity during the inference process. However, as opposed to guaranteed privacy provided by encryption-based algorithms, perturbation-based algorithms suffer from the possibility of privacy leakage, meaning sensitive private information may still be recoverable by the adversarial server despite the introduced perturbations.

He et al. (2019) presented one of the first systematic studies on model inversion attacks (MIA) on collaborative inference (CI). Their research shows that a shadow network can effectively emulate the client's secret network, enabling the recovery of raw images, especially when the client retains only one single convolutional layer. While the client is able to keep more privacy as it keeps more layers, such a method is less practical in the real world due to the limitations of the computational power of edge devices. Mireshghallah et al. (2020) proposed Shredder, which uses a noise injection layer before the client sending out computed results to reduce mutual information between client and server while maintaining good classification accuracy. Nevertheless, Lu et al. (2022) demonstrated that Shredder falls short in safeguarding facial images from recovery. In our own experimentation with the noise injection layer proposed by Shredder, applied to a ResNet-18 architecture on CIFAR-10, we observed significant accuracy drops with combined multiplicative and additive noise. On the other hand, simple additive noise resulting in approximately a 5 percent drop in accuracy failed to protect images from recovery, as depicted in Figure. 1. Lu proposed to use a policy-based processor between client and server to protect private information, but figures in their work seem to indicate that the effectiveness of their policy should be attributed to removing some regions from the original image that contain sensitive data. While such an approach is effective in some cases, it falls short in scenarios where sensitive information is embedded within the image, such as in facial authentication tasks.

In this paper, we aim to bring forth these contributions to the research community. Firstly, we expand upon the systematic analysis of various model split strategies between the client and server, focusing on more complex architectures commonly used in practice. Second, we take a different path from those approaches that propose different modifications to the data and introduce *Ensembler*, a secure collaborative inference framework designed to substantially increase the effort required to recover client input. *Ensembler* is not only a stand-alone framework that significantly increases the adversary server's reconstruction difficulty but can also be seamlessly integrated with existing complex algorithms to construct practical and secure inference architectures tailored to specific needs.

The remainder of this paper is organized as follows: Section 2 introduces the background of collaborative inference and related works, as well as formally defining the threat model. Section 3 offers a systematic analysis of the impact of different model split strategies on server-side reconstruction difficulty. Section 4 introduces *Ensembler* and details its design for better secure collaborative inference. Section 5 presents the empirical experiments related to *Ensembler* and showcases its effectiveness in protecting the client's private data, and Section 6 concludes the paper.

## 2 BACKGROUND

### 2.1 COLLABORATIVE MACHINE LEARNING

The development of mobile graphic processing units (GPUs) has ushered in a new era where machine learning tasks are increasingly deployed with a portion of the computation being handled by edge devices. Related areas include federated learning, where multiple edge devices jointly train a deep learning model (McMahan et al., 2017; Yu et al., 2023; Yaldiz et al., 2023); split learning, where a DL model is split into two or more parts, and the client and server jointly train it (Poirot et al., 2019); and collaborative inference, where a DL model is split, with only a portion deployed on the server to provide services (He et al., 2019; Osia et al., 2018). In this paper, we will focus on the inference part and assume that the training phase of DL models is secure. Though the training

phase is sometimes also susceptible to adversarial attacks aimed at stealing sensitive information (Inan et al., 2021; Li et al., 2022; Zhang et al., 2021), private inference is still more prevalent in most practical scenarios.

## 2.2 THREAT MODEL

In this paper, we consider the collaborative inference task between the client and the server, who acts as a semi-honest adversarial attacker that aims to steal the raw input from the client. Formally, we define the system as a collaborative inference on a pre-trained DNN model, $\mathbf{M}(x, \theta)$, where the client holds the first and the last a few layers (i.e. the "head" and "tail" of a neural network), denoted as $\mathbf{M}_{c,h}(x, \theta_{c,h})$ and $\mathbf{M}_{c,t}(x, \theta_{c,t})$. The rest of the layers of DNN are deployed on the server, denoted as $\mathbf{M}_s(x, \theta_s)$. $\theta$ is the trained weights of $M$, where $\theta = \{\theta_{c,h}, \theta_s, \theta_{c,t}\}$. The complete collaborative pipeline is thus to make a prediction of incoming image $x$ with $\mathbf{M}_{c,t}[\mathbf{M}_s[\mathbf{M}_{c,h}(x)]]$.

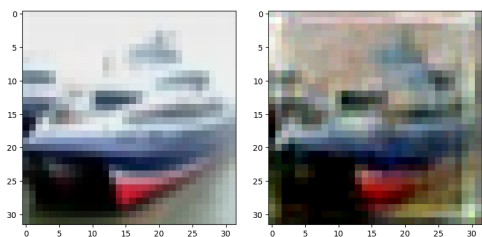

Figure 1: Sample image from CIFAR-10. Left is the original image and right is recovered image.

During the process, the server has access to $\theta_s$ and the intermediate output $\mathbf{M}_{c,h}(x)$. In addition, we assume that it has a good estimate of the DNN used for inference. That is, it has auxiliary information on the architecture of the entire DNN, as well as a dataset in the same distribution as the private training dataset used to train the DNN. However, it does not necessarily know the hyper-parameters, as well as engineering tricks used to train the model. Since the server is a computation-as-a-service (CaaS) provider, it is assumed to have reasonably large computation resources. While it is powerful in computing, the server is restricted from querying the client to receive a direct relationship between raw input $x$ and intermediate output $\mathbf{M}_{c,h}(x)$.

In order to reconstruct raw input $x$ from the intermediate output $\mathbf{M}_{c,h}(x)$, the server adopts a common model inversion attack (He et al., 2019; Lu et al., 2022; Dosovitskiy & Brox, 2016). It constructs a shadow network $\tilde{\mathbf{M}}(x, \tilde{\theta_{c,h}}, \theta_s, \tilde{\theta_{c,t}}) : \{\tilde{\mathbf{M}_{c,h}}, \mathbf{M}_{server}, \tilde{\mathbf{M}_{c,t}}\}$ such that $\tilde{\mathbf{M}}$ simulates the behavior of $\mathbf{M}$. After training $\tilde{\mathbf{M}}$, the adversarial server is able to obtain a representation $\tilde{\mathbf{M}_{c,h}}$ such that $\tilde{\mathbf{M}_{c,h}}(x) \sim \mathbf{M}_{c,h}(x)$. As the next step, with the help a decoder of $\tilde{\mathbf{M}_{c,h}}$ to reconstruct the raw image from intermediate representation, it is able to reconstruct the raw input from $\mathbf{M}_{c,h}(x)$.

## 2.3 ASSUMPTIONS OF OTHER RELATED WORKS

In this section, we provide an overview of various attack models and the assumptions adopted in other works related to collaborative inference (CI) under privacy-preserving machine learning (PPML). Since different works potentially use different collaboration strategies between the client and the server, we will use the generic notation, where $\mathbf{M}_c$ is held by the client, and $\mathbf{M}_s$ is held by the server. Generally, the attacks from the server will fall into three categories:

- **Training Dataset Reconstruction Attacks** that try to predict if certain attributes, including but not limited to individual samples, distributions, or certain properties, are a member of the private training set used to train $\mathbf{M}(x, \theta)$. If successful, the privacy of the training dataset will be compromised. We refer readers to the survey by Hu et al. (2022) and Salem et al. (2023) for more details.

- **Model Inversion Attacks** that try to recover a particular input during inference when its raw form is not shared by the client. For example, in an image classification task, the client may want to split $\mathbf{M}$ such that it only shares latent features computed locally to the server. However, upon successful model inversion attacks, the server will be able to generate the raw image for classification tasks based on the latent features. It is important to note that, in this paper, we adopt the same definition of model inversion attacks as of (He et al., 2019). This term also refers to attacks that reconstruct the private training dataset in other works. We will focus on reconstructing private raw input for the rest of the paper.

- **Model Extraction Attacks** that try to steal the parameters and even hyper-parameters of **M**. This type of attacks compromise the intellectual property of the private model and are often employed as sub-routines for model inversion attacks when the server lacks direct access to **M**'s parameters.

Different works also make different assumptions on the capability of the server. First, it is widely-accepted that the server has sufficiently yet reasonably large computing power and resources, as its role is often providing ML service. Regarding the auxiliary information on **M**, they generally fall into three levels:

- **White Box** assumes that the server has full access of architecture details of **M** such as the structure and parameters (Liu et al., 2021). Different definitions also add different auxiliary information available to the server, such as training dataset Liu et al. (2021), corrupted raw input Zhang et al. (2020), or a different dataset Wang & Kurz (2022). This setting is often associated with attacks that try to reconstruct private training dataset (Wang & Kurz, 2022; Zhang et al., 2020; Haim et al., 2022).

- **Black Box** assumes that the server does not have any information of neither **M** nor training dataset. However, it is allowed to send unlimited queries to the client to get $\mathbf{M}_c(x)$ (Xu et al., 2023; Kahla et al., 2022).

- **Query-Free** restricts the server from querying $\mathbf{M}_c$. While such an assumption greatly limits the reconstruction ability of the adversarial party, there are no limitations on auxiliary information available to the server besides the actual weights of $\mathbf{M}_c$. (He et al., 2019; Ding et al., 2023) have both shown that $\mathbf{M}_c$ is still vulnerable of leaking private information of the raw input when the server has information of the model architecture and training dataset. Our work will adopt this setting.

## 3 ANALYSIS ON SPLITTING BETWEEN CLIENT AND SERVER

Previous work from He et al. (2019) provided a systematic analysis on the recovery difficulty and quality of the above mentioned model-inversion attack. Their work analyzed effects on reconstruction quality from loosening assumptions of auxiliary information available to the server (DNN architecture and training dataset), as well as choosing different split points (h) between the client and the server. However, their work was based on a simple 6-layer convolutional neural network (CNN), which is seldom used in today's service. In this section, we further their analysis upon more practical architectures, namely ResNet-18 and VGG-16.

One of the important findings from He et al. (2019); Ding et al. (2023)'s study is that increasing the depth (h) of $\mathbf{M}_{c,h}$ will lead to worse image reconstruction quality of the adversarial attacker in MIA. At the same time, part of Zhou et al. (2022)'s algorithm lets the client, instead of the server, compute the Softmax function of $\mathbf{M}(x, \theta)$ at the last layer. The success of their algorithm raises the possibility of utilizing a second split point to enhancing privacy protection. Under the threat model defined by Section 2.2, we provide visual evaluations of the quality of reconstructed images from MIA, as shown by Fig. 2 and 3. The vertical axis is the position of first split point, and the horizontal

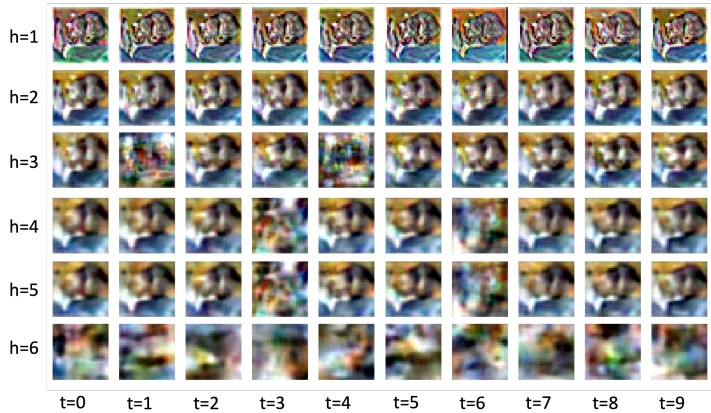

Figure 2: Effect of first and second split points on VGG16. The vertical axis is the first split point in terms of layers, and the horizontal axis is the second split point counting backwards on layers.

axis is the position of second split point counting backwards. For VGG-16 architecture, the first h layers of $\mathbf{M}$ belongs to the client. For the ResNet-18 architecture with 4 blocks, h represents the number of residual blocks computed by the client, with h=1 being the client only computing the first convolutional layer.

As shown in the figures, our experiments align with the results from He et al. (2019) and Ding et al. (2023). **The deeper the first split point is, the worse the reconstructed image is.** However, the experiments do not support the idea from Zhou et al. (2022). The second split point does not increase the difficulty of reconstruction under MIA. It is also noteworthy to point out that while our experiments indicate that image reconstruction quality is below human-level recognition after h=6 for VGG-16 and h=2 for ResNet-18, this should not be treated as a privacy-guarantee. This is because we are using a standard decoder

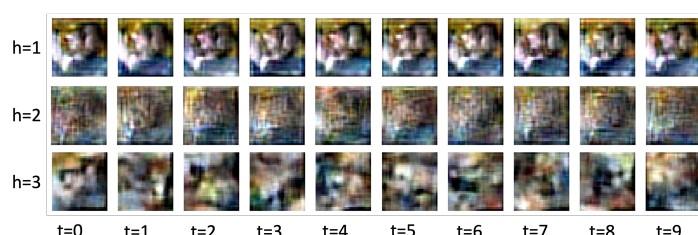

Figure 3: Effect of first and second split points on ResNet-18 with 4 blocks in the shape of [2,2,2,2]. The vertical axis is the first split point, and the horizontal axis is the second counting backwards. When h=1, only the first convolutional layer belongs to the client; when h=2, split point is at the end of the first block; when h=3, the split point is at the second block.

for $\tilde{\mathbf{M}}_{c,h}(x, \tilde{\theta}_{c,h})$, whereas there exist more powerful generative decoders that could do potentially better at reconstructing images (Khosravy et al., 2022). At the same time, this reconstruction depends on the task. For example, Lu et al. (2022) is able to reconstruct high-quality facial images with larger h, and Ding et al. (2023) is more successful with vehicle reconstruction. We also provide a brief experiment of MIA on NLP task in the Appendix A.1.

## 4    *Ensembler* ARCHITECTURE

While it is possible to protect sensitive data via increasing the depth (h) as shown by the previous section, such depth is often impractical for edge devices due to the computational demands involved. In this section, we present *Ensembler*, a framework that augments the privacy of intermediate information sent by the client without requiring extra computation efforts of the client during inference. *Ensembler* is highly extensible, and it is compatible with existing works that apply noises and perturbation during both DNN training and inference. We will go over the detailed architecture in Section 4.1, as well as the training stage of this new framework in Section 4.2.

### 4.1    ARCHITECTURE OVERVIEW

As illustrated in Fig. 4, *Ensembler* leverages model ensembling on the server to generate a regularized secret $\mathbf{M}_{c,h}$ that is hard to be reconstructed by the server. It consists of three parts: standard client layers, N different server nets, and a selector. During the collaborative inference pipeline, the client computes $\mathbf{M}_{c,h}(x)$ and transmits the intermediate output to the server. The server then feeds the intermediate output through each of the $\mathbf{M}_s^i$, and reporting the output of each $\mathbf{M}_s^i$ to the client. The client then employs a selector to perform a selection of the feedback from the server, which activates results of P out of N nets and combines them. As a final step, it performs the computation of the last t layers to classify the input. We will introduce these separately in this section.

### 4.1.1    CLIENT LAYERS

During collaborative inference, a part of the DNN is run by the client. Under the proposed framework, the client is responsible for running the first h layers $\mathbf{M}_{c,h}$ and the last t layers $\mathbf{M}_{c,t}$. These layers are the same as the client part of a typical collaborative inference framework. $\mathbf{M}_{c,h}$ takes the raw input (often an image) and outputs the intermediate result, whereas $\mathbf{M}_{c,t}$ takes the output from the serveras input and outputs the likelihood of each class.

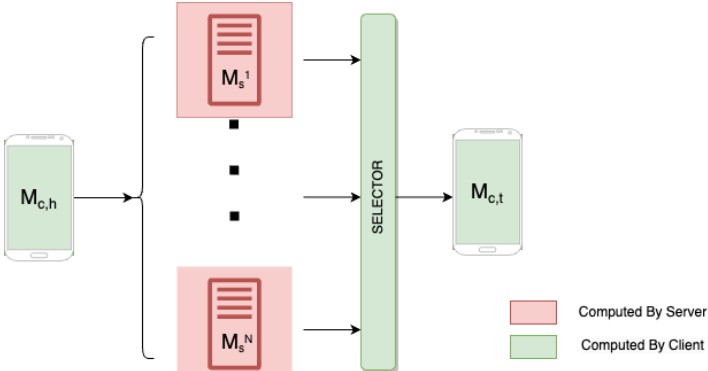

Figure 4: Illustration of the proposed architecture, *Ensembler*. Different from the traditional CI pipelines, it deploys N neural networks on the server, and uses a selector to activate P of the N nets.

### 4.1.2 SERVER NETS

On the server side, the network is consisted of N copies of DNN, with each $M_s^i$ corresponding to what the server would normally process in a typical collaborative inference pipeline. That is, each $M^i : \{M_{c,h}, M_s^i, M_{c,t}\}$ is a valid pipeline for the inference task. Upon receiving $\mathbf{M}_{c,h}(x)$, which is the input from the client, the server shall feed this input into each of $M_s^i$, and it outputs N representations of hidden features used for classification.

### 4.1.3 SELECTOR

To increase the difficulty of the server reconstructing the model and recovering the raw input, a selector is applied before the last layer run by the client. The selector serves as a secret activation function, which activates P of the N nets according to Equation. 1, where $S_i$ is the activation from selector, and $\odot$ is the element-wise multiplication. For simplicity, we consider $S_i = 1/P$ if $M_s^i$ is selected by the client, an $S_i = 0$ otherwise.

$$Selector[M_s(x)] = Concat[S_1 \odot M_s^1(x), S_2 \odot M_s^2(x), ..., S_N \odot M_s^N(x)] \tag{1}$$

### 4.2 TRAINING STAGE

As mentioned above, the design choices of *Ensembler* aim to achieve a regularized $M_{c,h}$ such that a shadow network based on $M_s$ would be an incorrect estimate of $M_{c,h}$. To achieve this goal, the proposed architecture uses a two-staged training pipeline. For the first stage, it needs to obtain N distinct $M^i(x, \theta^i) : \{M_{c,h}^i, M_s^i, M_{c,t}^i\}$ such that a shadow network that accurately simulates $M_{c,h}^i$ could not simulate $M_{c,h}^j$. In our approach, we choose to simply introduce a Gaussian noise layer after the intermediate output $M_{c,h}^i(x)$. The objective function in this stage is to minimize the cross-entropy loss in the form of Equation. 2, where $N(0, \sigma)^i$ is a fixed Gaussian noise added to the intermediate output. The choice of $\sigma$ is dependent on the quality of training, and given the inherent redundancy in the parameters of DNNs, adding some noises will not affect the classification accuracy. For example, adding noise of $N(0, 0.1)$ after the first layer of a ResNet-18 architecture for CIFAR-10 image classification task results in less than 1% accuracy loss. We choose Gaussian noise because of simplicity in implementation, and we'd argue that any method that will lead to distinctive $M_{c,h}$ will be sufficient for this step. However, this step is nonetheless needed to ensure that each model has different parameter weights. Otherwise, all N models would be identical to each other, and the framework fails its purpose in protecting privacy.

$$L_\theta^i = -\sum_j y_j * log\, M_{c,t}^i(M_s^i[M_{c,h}^i(x) + N(0, \sigma)^i])_j \tag{2}$$

After the first training stage, N different DNNs are obtained. The proposed framework selects P of the N nets, and retrains an "ensembled" network, *Ensembler*, which has been outlined in the previous section. During the training, parameters of $M_s$ are frozen. This step is used to ensure the performance of the model during inference. While the training process is just like any typical neural network, it is noteworthy to point out that we add a regularization term to the standard cross-entropy loss to enforce $M$ to learn a joint $M_{c,h}$ and $M_{c,t}$ representation from all of the P server nets. The custom loss function, as shown in Equation. 3, adds a high penalty to the model if the gradient descends only to the direction of some single server net $M_s^i$. In the equation, CS is the cosine similarity, and $\lambda$ is a hyper-parameter controlling the regularization strength. Since this is an end-to-end training process, any perturbation-based algorithms could be seamlessly combined with the proposed framework during this step to provide further privacy protection. For our experiment, we just choose simple Gaussian noises to be consistent with the first step.

$$L_\theta = -\sum_i^{i \in N} \sum_j [y_j * log \, M_{c,t}(Selector[M_s^i[M_{c,h}(x) + N(0,\sigma)]])_j] \tag{3}$$
$$+\lambda max_{i \in P}[CS(M_{c,h}(x), M_{c,h}^i(x))]$$

### 4.3 Intuition behind *Ensembler*

In this section, we discuss the intuition behind the proposed architecture. Since the attacker will construct shadow networks to simulate the behavior of client's private networks, the exact purpose of the two-staged training algorithm is to ensure that the attacker is not able to learn the selector with its shadow network. Through the first stage of training, N different models that have distinctive weights are obtained, yet all of them are able to make comparative predictions on the dataset. An arbitrary ensemble of P out of the N networks will form a new network, whose $M_{c,h}$ will be distinctive from networks under a different combination. That is, since $M_s^{i+j}$ would be different from $M_s^{i+k}$, $M_{c,h}^{i+j}$ obtained from $M_s^{i+j}$ would be different from $M_{c,h}^{i+k}$ obtained from $M_s^{i+k}$, where $+$ is ensemble of server nets. Thus, with N networks in the first stage of the algorithm, we will have $2^N$ different possible $M_{c,h}$ that could be the valid answer to the shadow network. When the attacker tries to train an adaptive attacker, the shadow network will learn an arbitrary representation $\tilde{\mathbf{M}_{c,h}}$ and an arbitrary $\tilde{S}$. Such combination is a valid choice in terms of classification accuracy but is nonetheless incorrect compared to the actual $\mathbf{M_{c,h}}$.

### 4.4 Time complexity of *Ensembler*

From previous section, it is clear that the time complexity of the proposed framework is N times of the individual network on a single-core GPU, and there is negligible extra communication cost between the client and the server. However, it is worthy to emphasize that since each $M_s^i$ is independent to each other, the proposed framework is friendly to parallel execution and even multiparty (multi-server) inference. Under those settings, the theoretical time complexity of N would be replaced with lower practical time costs or even causes the framework to be uninvertible. On the other hand, since the server is not able to adaptively learn the client's representation, the only option is to exhaustively try all combinations, which takes $2^N$ times compared to reconstructing a single network. Here, we provide a semi-formal argument on exponential complexity of reconstructing the best quality image under *Ensembler* protection.

**Lemma 1** Reconstructing image from single neural network $\mathbf{M}_s^i$ is not viable.

For any shadow network obtained through single $\mathbf{M}_s^i$ , it needs to first simulate the behavior of $\mathbf{M}_{c,h}^i$. In this case, if there exists some $\tilde{\mathbf{M}}_{c,h}^i$ that simulates $\mathbf{M}_{c,h}$, the training loss of the second training phrase is not optimized (Equation. 3) due to the regularization term.

**Lemma 2** Reconstructing image from incorrect choice of $\mathbf{M}_{activated} = [\mathbf{M}_s^i, ..., \mathbf{M}_s^j]$ is not viable.

Since $\mathbf{g}_i \in N(0, \sigma)$ are independent of each other, the N different $M^i(x, \theta^i)$ obtained in the first training stage are also distinctive. Including incorrect $\mathbf{M}_s^j$ in the shadow network construction will lead to the model regularizing in an incorrect direction.

**Conclusion** The time complexity of reconstructing best quality input from N server nets is theoretically $2^N - 1$.

## 5 EXPERIMENTS AND EVALUATIONS

### 5.1 ARCHITECTURE DETAILS

During the experiment, we consider the most strict setting, where h=1 and t=1 on a ResNet-18 architecture for three image classification tasks, CIFAR-10, CIFAR-100, and a subset of CelebA-HQ Zhu et al. (2022). That is, the client only holds the first convolutional layer as well as the last fully-connected layer, which is also the minimum requirement for our framework. For CIFAR-10, the intermediate output's feature size is [64x16x16], for CIFAR-100, we remove the MaxPooling layer and the intermediate output's feature size is [64x32x32], and for CelebA, the intermediate output's feature size is [64x64x64]. We consider the ensembled network to contain 10 neural networks (N=10), each being a ResNet-18. The selector secretly selects {4,3,5} out of the 10 nets (P={4,3,5}), respectively. The adversarial server is aware of the architecture and the training dataset. It constructs a shadow network $\tilde{M}_{c,h}$ consisted of three convolutional layers with 64 channels each, with the first one simulating the unknown $M_{c,h}$, and the other two simulating the Gaussian noise added to the intermediate output. It also has $\tilde{M}_{c,t}$ with the same shape as $M_{c,t}$. For adaptive shadow network, it learns from all 10 server nets with an additional activation layer that is identical to the selector. For any noises added to the intermediate outputs during the training and inference stage, we consider a fixed Gaussian noise $g \sim N(0, 0.1)$.

### 5.2 EXPERIMENT SETUP

To evaluate the effectiveness of our approach, we employ three key metrics: Structural Similarity (SSIM), Peak Signal to Noise Ratio (PSNR), and visual assessment. The first two metrics offer quantitative evaluations of the reconstruction quality of MIA, with higher SSIM and PSNR values indicating better reconstruction quality. As our proposed architecture operates in parallel with existing perturbation methods, we consider the following baseline approaches for comparison on CIFAR-10: no protection (NONE), adding small noise in a single network that does not require retraining (Shredder Mireshghallah et al. (2020)), adding large noise and retrain a single network (Single), and adding dropout layer in the single network or ensembled network, but with only one round of training (DR-single and DR-ensemble). The dropout is included to differentiate our architecture with dropout layers, as the selector component does look very similar to a dropout layer. For the other two datasets, we select some of the important benchmarsk for comparison. For CelebA-HQ, since the intermediate output's feature size is too large for the simple Gaussian filter to be visually effective, we add an untrained random $M_{c,h}$ (Random) to illustrate the maximum capacity of Gaussian filter at the cost of accuracy. For the proposed architecture, we evaluate the performance of both reconstruction of a single neural network (N=1), as well as reconstruction using the entire network (Adaptive). For reconstruction of ensembled nets using a single neural network, we report the best reconstruction result of the N nets. For Section 3, we implement the experiments on a server with four A-6000 GPUs using Python and PyTorch. For Section 4, we used a mixture of the server and Google Colab, which uses one T4 GPU.

### 5.3 COMPARISON OF RESULTS

We provide the quantitative evaluations for CIFAR-10 in Table. 1, and the visual assessments in Figure. 5 in Appendix A.2.1. It could be seen that the proposed framework significantly increases the reconstruction difficulty of the adversarial party. *Ensembler* incurs 2.13% drop in classification accuracy compared to the model without any protection, which is marginal compared to its advantage in protecting privacy of the client's raw input. From the figure, it is clear that the reconstructed images are hardly recognizable by human-level interpretations.

In addition, we provide the quantitative evaluations for CIFAR-100 and CelebA-HQ in Table. 2 and 3, and the visual assessments in Appendix A.2.2 and A.2.3. The proposed framework remains effective when the feature size increases. In particular, the framework safegaurds the model's prediction ability while protecting the input images on par with the random head network. Although

the visual assessments show that increasing feature size leads to better visual recognition, we argue that it is inevitable with simple Gaussian noises. In particular, the shadow network is able to raise the reconstruction quality of a totally mismatched random $M_{c,h}$ to beyond human-recognition level from the shadow network with best PSNR.

Table 1: Quantitative evaluations of the different defense mechanisms with CIFAR-10. Last three are the proposed framework. For SSIM and PSNR, lower values mean worse reconstruction quality.

| Name | Change in accuracy | SSIM | PSNR |
|---|---|---|---|
| NONE | 0.00% | 0.4363 | 12.2678 |
| Shredder | -5.68% | 0.5359 | 10.4033 |
| Single | 2.15% | 0.3921 | 7.5266 |
| Dr-single | **2.70%** | 0.3453 | 6.6674 |
| Dr-ensemble (best SSIM) | 1.42% | 0.373 | 7.3493 |
| Dr-ensemble (best PSNR) | 1.42% | 0.3232 | 7.9598 |
| Adaptive | -2.13% | **0.0555** | **5.981** |
| N=1 (best SSIM) | -2.13% | **0.2889** | **4.865** |
| N=1 (best PSNR) | -2.13% | **0.2221** | **5.5348** |

Table 2: Quantitative evaluations of the different defense mechanisms with CIFAR-100. Last two are the proposed framework. For SSIM and PSNR, lower values mean worse reconstruction quality.

| Name | Change in accuracy | SSIM | PSNR |
|---|---|---|---|
| Single | -0.97% | 0.4558 | 8.5225 |
| Adaptive | **0.31%** | **0.0864** | **4.7715** |
| N=1 (best SSIM&best PSNR) | **0.31%** | **0.2636** | **5.0741** |

Table 3: Quantitative evaluations of the different defense mechanisms with CelebA-HQ Zhu et al. (2022). Last two are the proposed framework. For SSIM and PSNR, lower values mean worse reconstruction quality.

| Name | Change in accuracy | SSIM | PSNR |
|---|---|---|---|
| Single | -1.24% | 0.2650 | 14.3126 |
| Random (best SSIM&best PSNR) | -65.19% | **0.1387** | **12.8150** |
| Adaptive | **2.39%** | 0.0897 | 13.3698 |
| N=1 (best SSIM&best PSNR) | **2.39%** | 0.1791 | 12.0645 |

## 6 CONCLUSION

In this paper, we present two contributions to the research community of PPML and collaborative inference. First, we extend the discussion on choosing the split points between client and server under collaborative inference. Our experiments illuminate that deeper split points yield lower-quality reconstructions, while the introduction of a second split point offers little to no improvement. Furthermore, we introduce a novel framework, *Ensembler*, designed to significantly increase the complexity of reconstruction for adversarial parties. Ensembler seamlessly aligns with existing methods that introduce diverse forms of noise to intermediate outputs, potentially yielding robust and adaptable architectures if combined with them. Our experiments highlight the substantial deterioration in reconstruction quality for images safeguarded by Ensembler when compared to those without its protection.

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

# A APPENDIX

## A.1 MIA ON TEXT DATA

As large language models (LLM) develop in recent years, privacy of text data has also brought forth much attention. There have been several works applying MIA on LLMs, but they all at least require black-box query-access to $M_{c,h}$ (Song & Raghunathan, 2020; Li et al., 2023). To the best of our knowledge, MIA for text data under a query-free threat model has not been studied.

Table 4: Example of English to German Translation. From left to right, the columns are: 1. ground truth of raw input in English. 2. translated text in German during collaborative inference. 3. translated text in German using shadow network as the model. 4. decoded input from the intermediate output of shadow network (to ensure the decoder is working properly). 5. decoded input from the intermediate output produced by $M_{c,h}$

| Ground Truth | Standard Translation | Shadow Translation | Decoder On Shadow | Decoder On Client |
|---|---|---|---|---|
| There is a young girl on her cellphone while skating. | Ein junges Mädchen fährt mit ihrem Handy während sie skatet . | Ein junges Mädchen fährt auf ihrem Handy und telefoniert dabei , während sie auf ihrem Handy fährt | There is a young girl skating while skating on her cellphone . | Dark swinging complete wait for no complete complete complete wearing complete wait wait |
| Trendy girl talking on her cellphone while gliding slowly down the street | Ein Mädchen mit Glatze spricht auf ihrem Handy , während sie die Straße entlangfährt . | Ein Mädchen spricht auf der Straße , während sie die Straße spricht . | gliding girl talking on her cellphone while gliding slowly down the street | Here picture waves balding wait for a picture complete wearing picture wait mixing wait mixing wait mixing wait |

In the experiments, we tried to use the same MIA pipeline from the image reconstruction to text reconstruction by considering an example model from PyTorch [1]. This is a transformer model for English to German translation tasks, and for collaborative inference, the client privately holds the embedding layer for the two languages. The text results for reconstruction are summarized in Table. 4, which is a standard MIA pipeline from left to right. The second column showcases the translated text under normal inference. The adversarial server trains a shadow network, whose translation is shown in the third column. From the shadow network, a decoder is trained to reconstruct the input from intermediate output. Finally, the decoder is applied to the intermediate output from the client, with the results illustrated in the last column. As it suggests on the last two columns, while the decoder is able to reconstruct text on the shadow network $\tilde{M}_{c,h}$, due to the moderate difference between $\tilde{M}_{c,h}$ and $M_{c,h}$, the decoded result from $M_{c,h}$ could not be understood. It is also noteworthy to mention that the shadow model is much harder to converge compared to training the entire $M$, possibly because majority of parameters are frozen in the network.

---

[1] `https://pytorch.org/tutorials/beginner/translation_transformer.html`

## A.2 IMAGES

### A.2.1 VISUAL ILLUSTRATIONS FOR CIFAR-10

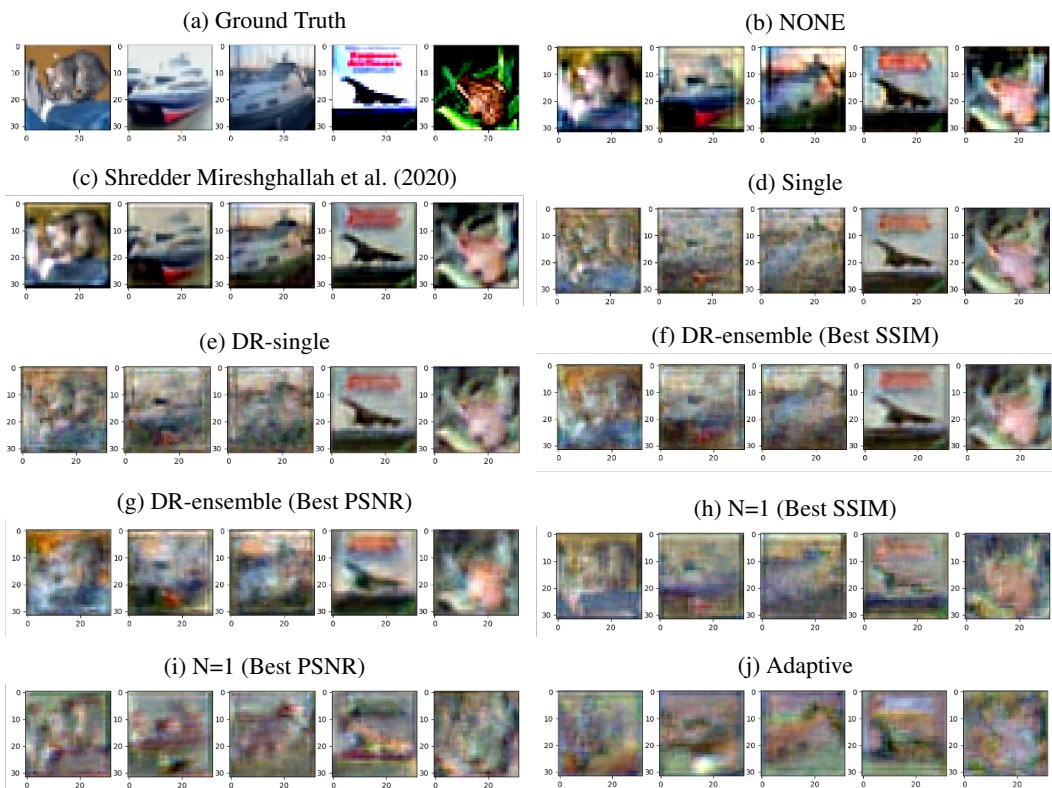

Figure 5: Visual Assessment of the image reconstruction quality under different defense mechanisms for CIFAR10. It is clear that our proposed framework (h,i,j) is effective. The effect is more obvious on the fourth image.

### A.2.2 VISUAL ILLUSTRATIONS FOR CIFAR-100

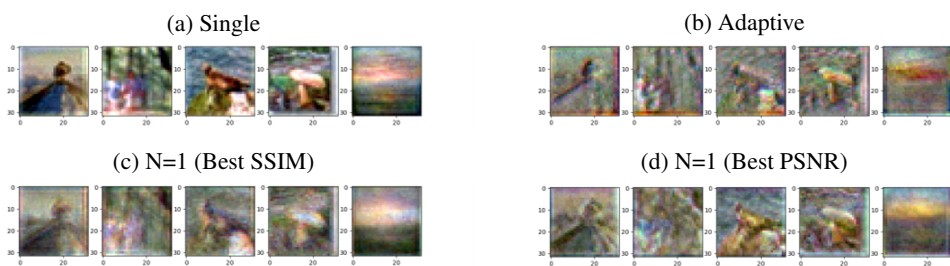

Figure 6: Visual Assessment of the image reconstruction quality under different defense mechanisms for CIFAR-100.

### A.2.3 VISUAL ILLUSTRATIONS FOR CELEBA-HQ

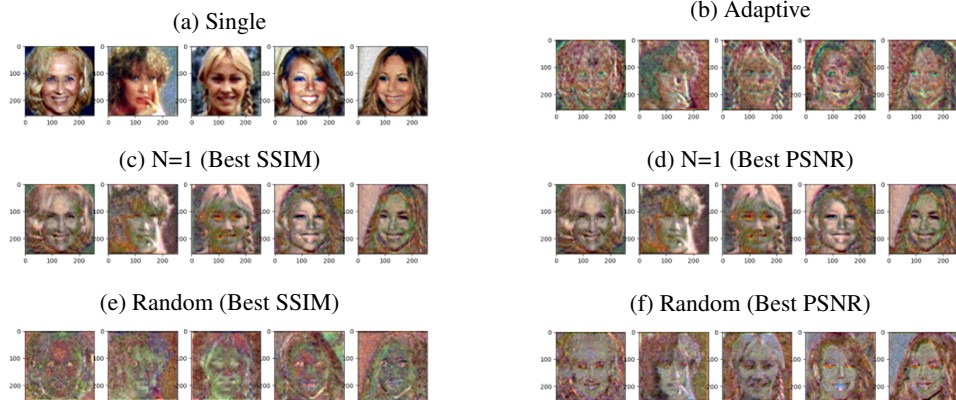

Figure 7: Visual Assessment of the image reconstruction quality under different defense mechanisms for CelebA-HQ. When constructing shadow networks and decoders using single pipeline, the decoder that results in best PSNR and SSIM is the same one.

