# OpenReview forum: "Ensembler: Combating model inversion attacks using model ensemble during collaborative inference"
_ICLR.cc/2024/Conference — Submitted to ICLR 2024_

### Official Review · Reviewer_H7PD · 2023-10-29

**Soundness:** 3 good
**Presentation:** 3 good
**Contribution:** 3 good
**Rating:** 5
**Confidence:** 2

**Summary:**

In this paper, the authors introduce Ensembler, an extensible framework designed to substantially increase the difficulty of conducting model inversion attacks for adversarial parties. Ensembler leverages model ensembling on the adversarial server, running in parallel with existing approaches that introduce perturbations to sensitive data during colloborative inference. Their experiments demonstrate that Ensembler can effectively shield images from reconstruction attacks, achieving recognition levels that fall below human performance in some strict settings.

**Strengths:**

- Propose Ensembler, an extensible framework designed to substantially increase the difficulty of conducting model inversion attacks for adversarial parties
- Comprehensive evaluation on Ensembler

**Weaknesses:**

- The threat model is not compelling

The threat model in this paper is not compelling. The authors claim that "it has auxiliary information on the architecture of the entire DNN, as well as a dataset in the same distribution as the private training dataset used to train the DNN". In real world, it is unclear how to obtain the information for the attackers. Also, the authors assumed the server has reasonably large computation resources but no detailed discussion or analysis on it. Thus, it is unknown how practical it is for the threat model. Without valid threat model, it is unclear the usefulness of this paper.

- Lack of practicality

The practicality of implementing "Ensembler" in a real-world scenario is somewhat unclear, given the inherent complexity and computational overhead of maintaining multiple models and the distillation process. Also, this paper mainly target on small-scale image dataset such as CIFAR-10. It is unclear whether their findings or method can be generalized to large-scale settings. Thus, the practicality is damaged.

**Questions:**

Need to improve or provide justification on the threat model and practicality.

---

> ### Author Response · Authors · 2023-11-21
>
> Dear reviewer,
>
> We want to thank you for your valuable suggestions, and we try to address your concerns here.
>
> The reviewer questions on the threat model and practicability of the proposed framework.
>
> For threat models, we are not able to comment on how to get those auxiliary information. However, when considering the possible attacks and the related defenses, it is always good to consider the extreme cases. This is why we assume that 1. Server somehow guesses the dataset and model architecture correctly. 2. The client only holds 1 layer for head and 1 layer for tail. Actually, the first assumption is not as hard as it seems. For the model architecture, since the client has very limited computing resources, the server could just fill the top layers with convolutional layers, and last layers with dense layers. It has a “high chance” of getting it correctly. For the dataset, the server could just use the standard open-sourced training datasets. Since it has large computing resources, it could afford the price to train on a few of them, which could match the one used by the client. Of course, if the server guesses incorrectly, then this is to the advantage of the client as the reconstruction quality will be much worse, as shown by [1].
>
> We have added some experiments on larger scale datasets. Please see the response to all reviewers for details.
>
>
> [1] Zecheng He, Tianwei Zhang, and Ruby B. Lee. Model inversion attacks against collaborative inference. In Proceedings of the 35th Annual Computer Security Applications Conference, ACSAC ’19, pp. 148–162, New York, NY, USA, 2019.

---

### Official Review · Reviewer_oZxB · 2023-10-31

**Soundness:** 1 poor
**Presentation:** 2 fair
**Contribution:** 1 poor
**Rating:** 3
**Confidence:** 4

**Summary:**

This paper proposes a framework an ensembling-based approach to defend against reconstruction attacks during collaborative inference. Ensembler (the proposed technique) uses model ensembling on the server and a secret selector (activations to compute weighted sums of model activations) on the client to prevent the server from reconstructing the raw input from the intermediate output. The paper also analyzes the effects of different model split strategies on the reconstruction difficulty and quality.

**Strengths:**

- The analysis around choice of split point, and how it impacts reconstruction, is insightful.
- The regularization in Eq 3. is well motivated, and the choice of simple noise addition (instead of some over-engineered, overly complex method) is nice.

**Weaknesses:**

My biggest issue is contrived threat model in this work. The server is assumed to be potentially malicious (which is reasonable) but at the same time, the proposed defense requires the server to train multiple models. If the server is indeed malicious, why would it want to do so? Even if it does, how can the client know if the server is malicious in its part of the models as well. For instance, it could shuffle or modify the outputs of its server nets such that only some of them are useful for the client model.

On that note, if the server is indeed malicious, a very simple attack can be: server sends N queries to the client posing as a user, observes intermediate outputs of client's model (sent to it normally), and then also observes the final output posing as the user. This can make reconstruction attacks (perhaps via shadow models) much more potent.

Additionally, the client is supposed to be computationally restricted, which is why a server is involved in the first place. How, then, is the client able to train the entire model locally (Section 4.2)? If the client is indeed powerful enough to train models (which requires at least as much memory as inference), then why not just run the model locally without any server?

## Minor comments

- Section 1, Pg2 "...adds privacy to the DL pipeline". What you really mean to say is that "current attacks are harder to run successfully on proposed technique", which is not the same as claiming increased privacy.

- Section 2.3 is missing the threat model of property/distribution inference. There are some works specifically for collaborative learning as well [1]. See [2] for a summary of this (and other) privacy risks in ML.

- Section 3.1, Pg5 "...our experiments align with the..." please avoid jumping to results before explaining the experimental setup.

- Section 3.2 does not seem to add any value- please consider removing it, and instead using that space to better explore results towards the end of the paper.

- Section 4.1 "As illustrated in Fig. 4..." please keep referenced figures on the same page as the first reference (ideally close to the reference, but at least same page).

- Section 4.1 "...the client secretly computes" - this is just normal computation on the client side, and there is nothing explicitly "secret" about it apart from the fact that the server does not see these computations, which is true by design irrespective of any techniques proposed in this work.

- Equation 1: $S_i$ is an activation, but what is the input to the activation? Please clarify.

- Section 5.3 is all I see for analysis of results. for a 9 page paper, there should definitely be much more space reserved for analyzing results.

### References
- [1] Zhang, Wanrong, Shruti Tople, and Olga Ohrimenko. "Leakage of dataset properties in Multi-Party machine learning." USENIX, 2021
- [2] Salem, Ahmed, et al. "SoK: Let the privacy games begin! A unified treatment of data inference privacy in machine learning." IEEE S&P, 2023

**Questions:**

- Section 2.2 "...where the client holds the first and last few layers" - are such splits standard practice? If yes, cite references clearly to works that have similar setups. If not, make that clear and justify why this split design makes sense.

- Section 2,2 "...a good estimate of the DNN used for inference..." - why is it just a good estimate? Isn't the server the party that usually trains the entire network to begin with?

---

> ### Author Response · Authors · 2023-11-21
>
> Dear reviewer,
>
> We want to thank you for your valuable suggestions, and we try to address your concerns here.
>
> The reviewer’s major concern seems to be on the threat model. We want to apologize for any confusions, but also to point out that our threat model targets specifically on collaborative inference. We made it clear in Section 2.1 that we assume the training phase is secure. This is a reasonable setting because training and deployment have different pipelines. For training of ML models, individual and small companies could buy a single GPU and train on their own machine. This is a one-time cost, and it is usually not as expensive. But deployment of ML service is continuous, which often requires a cloud server (which could be malicious). Concretely, if I want to build an image-classification app with privacy features, I could buy one GPU and train my model locally. Then, I would rent a cloud server that offers computation, and connects the app to the cloud. In this case, the app serves as the client, and the cloud is the malicious server.
>
> The reviewer also asks about the server shuffling and modifying the server nets. We want to strengthen that we consider a semi-honest adversary attacker. The definition of semi-honest attacker is that the semi-honest party has to follow the predefined protocols. It can curiously seek extra information, but it cannot change the pipeline.
>
> Finally, the reviewer asks about the server sending queries to the client. This is called a “black-box” attack. We have provided discussion on works related to this area. We also want to point out that our work focuses on “query-free” attacks, which explicitly prohibits the server querying the client to get the direct relationship between input and intermediate output. Concretely, using the example I gave above, unless the server is a member of the app users, it will not have access to querying the service. How to protect the adversary server becoming a member of the app is out of the concern of this paper, but a possible suggestion is to take extra care when renting the cloud service so that the server is not able to trace back to the app.
>
> The reviewer also raises some minor comments. We want to express our gratitude to those comments, and have modified the paper based on those comments.

---

> > ### Comment · Reviewer_oZxB · 2023-11-22
> >
> > If a cloud service is indeed used: most services that provide containers/dockers are indeed isolated, and there is no way for a server to be "malicious" in terms of the usage. The threat model still seems forced.  The fact that the threat model also forces the attacker to be "query free" makes it a highly niche threat model, which is not of much interest or practical relevance.
> >
> > I am not convinced with the authors' responses and am choosing to keep my score.

---

### Official Review · Reviewer_mcWL · 2023-11-01

**Soundness:** 1 poor
**Presentation:** 1 poor
**Contribution:** 2 fair
**Rating:** 3
**Confidence:** 4

**Summary:**

This work wants to propose a defense against model inversion attacks during collaborative inferences.

For collaborative inferences, a part of the forward passes (typically corresponding to the layers in the middle) is computed by the cloud server, and model inversion attacks in the context of collaborative inferences consider adversarial servers who attempt to reconstruct the original inputs.

They first empirically evaluate how the model splitting strategy (i.e. how many first&last layers are kept locally without sharing to the cloud server) affect the effectiveness of model inversion attacks. They evaluate this on VGG16 and ResNet-18 for image data (CIFAR-10), and they evaluate on an English to German translator for text data. They confirm previous results that keeping more first layers locally in the client side (i.e. not sharing them to the adversarial server) reduce the effectiveness of model inversion attack.

Secondly, they want to propose a defense against model inversion during collaborative inference that does not increase the local computation at inference time. They do so by training N different models, keeping their middle layers and then training a new pair of first layers&last layers with a regularization term encouraging it to utilize P of these middle layers. Here N and P are both pre-defined hyper-parameters. They evaluate this defense on CIFAR-10.

**Strengths:**

1. The topic is meaningful. Collaborative inference is a practical case and it is important to understand how big the threat from model inversion attacks can be and if one can mitigate the threats.
2. The idea of their defense looks novel.

**Weaknesses:**

1. **The experiments are really insufficient** in a sense that the evaluated scenario is limited (e.g. only evaluating on CIFAR-10 with ResNet-18)!
In section 3, when studying how model splitting strategies affect the effectiveness of model inversion attacks, only a single qualitative result is shown in each case (a total of 3 samples, one for each of VGG16, ResNet18 and English to German translator); In section 5, the proposed Ensembler defense is evaluated only on CIFAR-10 with ResNet-18 architecture.

2. **The proposed defense is neither well-explained nor well-investigated.**
The motivation behind the proposed design is not clear: Why/How would this increase the difficulty of model inversion attacks?
(1) Intuitively, each of the N group of middle layers obtained by the server are intentionally biased during training (through adding a fixed perturbation to its input). Since it is assumed that the attacker has access to a dataset with similar distribution, is it possible that such bias can be reversed by adding another learnable perturbation to the input of these middle layers? (2) Since the actual model uses only P out of N models, could the attacker also learn to select a subset of models that maximize the reconstruction quality?

**It is important to have an adaptive attack** by incorporating both (1) and (2)  in evaluating the proposed defense (unless one formally proves its effectiveness). Based on current supports from the paper, the proposed defense offers only complexity but not security.


3. **Missing discussion regarding the cost of the proposed defense.**
Another concern regarding the proposed defenses is the added overhead, both to the training phase and to the inference, but this is not well discussed in the paper.

**Questions:**

Please also refer to the Weakness part for my concerns regarding this paper (I would say priority-wise: Weakness 2 > Weakness 1 ~ Weakness 3, but they are all important issues).

Specifically:
1. For Weakness 1, I would suggest authors to include quantitative results for section 3 and to include evaluations on another dataset with higher resolution (e.g. some subsets of ImageNet or even tiny-ImageNet would be much more convincing than only having results from CIFAR-10). It would also be better to have results on more architectures but having only CIFAR-10 results is a bigger issue.

2. For Weakness 2, as I wrote above: (1) Intuitively, each of the N group of middle layers obtained by the server are intentionally biased during training (through adding a fixed perturbation to its input). Since it is assumed that the attacker has access to a dataset with similar distribution, is it possible that such bias can be reversed by adding another learnable perturbation to the input of these middle layers? (2) Since the actual model uses only P out of N models, could the attacker also learn to select a subset of models that maximize the reconstruction quality? **It is important to have an adaptive attack** by incorporating both (1) and (2)  in evaluating the proposed defense (unless one formally proves its effectiveness).

3. For Weakness 3, please let me know if you have the discussion somewhere but I miss it. Otherwise, I would recommend to include the discussions, even if it reveals some limitations.

---

> ### Author Response · Authors · 2023-11-21
>
> Dear reviewer,
>
> We want to thank you for your valuable suggestions, and we try to address your concerns here.
>
> Weakness1: The experiments are really insufficient
>
> The reviewer wants to see more experiments on both Section 3 and 5. Due to time and resource limits, we are not able to expand Section 3. However, Section 3 serves mainly as a complementary component to our proposed framework. All we want to say in Section 3 is that the conclusion drawn from previous papers is correct. “The deeper the first split point is, the worse the reconstructed image is.” On the other hand, the introduction of a second split point has no benefits to the defense. Section 3 will help us eliminate some of the concerns on the usage of two split points in our framework. While this is a necessity for Ensembler, it is seldom used before, making some people wonder if the increased protection should be attributed to this second split point.
>
> As discussed in the response to all reviewers, we have included more experiments in Section 5. We want to thank the reviewer for giving very detailed suggestions on using ImageNet or Tiny-ImageNet. We tried to use Tiny-ImageNet, but due to resource constraints, we anticipate that we will not be able to finish the experiments by the deadline. Thus, we consider the two datasets. CIFAR100 has equivalent intermediate feature size, using a modified network, with Tiny-ImageNet; and CelebA-HQ has similar intermediate feature size with ImageNet. Due to resource limits, we could not consider more architectures, but we do believe that the proposed model is agnostic to architecture choices.
>
> Weakness2: The proposed defense is neither well-explained nor well-investigated.
>
> We address the intuition of the proposed framework in the response to all reviewers. The reviewer asks two specific questions, and we will address them here:
> Is it possible that such bias can be reversed by adding another learnable perturbation to the input of these middle layers?
> Yes. In fact, the Gaussian noise could easily be approximated by a few convolutional layers. This is what we did for the shadow network. For Gaussian noise, we use 2 convolution layers with kernel size of 3 and 64 filters to approximate it. From single/Dr-single, we see that it works well.
> However, the whole purpose of the Gaussian kernel in the first stage of training is to ensure that each network, especially the server nets, has distinctive weights, yet remaining good accuracy. The key of the proposed framework is not about Gaussian noise. Rather, it is an algorithm that generates 2^N potential correct answers with N times complexity.
> Could the attacker also learn to select a subset of models that maximize the reconstruction quality?
> We address this in the response to all reviewers. Each possible combination of server nets would result in a different $M_{c,h}$, which gives good classification accuracy. However, only one of them is the “correct” one that maximizes the reconstruction quality. The shadow model is not able to learn the reconstruction quality because it does not know which one is the “correct” one.
>
>
> Weakness3: Missing discussion regarding the cost of the proposed defense.
>
> We address this one in the response to all reviewers, in Major concern 2.a, second paragraph.

---

> > ### Comment · Reviewer_mcWL · 2023-11-22
> >
> > Sorry I didn't get a chance to look at the rebuttals carefully as I have been rather occupied in the last a couple days.
> >
> > I will take a close look at your responses later and make sure to work with other reviewers towards a well-informed decision.
> >
> > However, after a quick and preliminary scan, a concern is the claim that, and I quote, 'only one of them is the “correct” one that maximizes the reconstruction quality. The shadow model is not able to learn the reconstruction quality because it does not know which one is the “correct” one.'
> >
> > While it is totally fine that you use this as an intuition, it is not a strong support to whether or not the proposed defense can be effective, because it may not be necessary for attackers to identify the exact subset of models being selected in order to recover the inputs to some extents, as suggested by the 'visual illustrations' included in Appendix (where one can clearly see the contours of the original images in the reconstruction).
> >
> > But again, I will take a closer look later and incorporate the opinions of other reviewers.

---

> > > ### Author Response · Authors · 2023-11-22
> > >
> > > Dear reviewer,
> > >
> > > Thank you very much for your response. You concern is totally valid, and indeed, "it may not be necessary for attackers to identify the exact subset of models being selected in order to recover the inputs to some extents." Still, by the construction of regularization in the loss function, the server needs to select the correct subset of server nets for "maximum" reconstruction quality.
> > >
> > > Regarding the reconstruction quality of the two new datasets, we think that the contours appearing in the reconstructed images for CelebA and in some images of CIFAR-100 are due to the limitations of simplicity of Gaussian noise and the client only holding one layer of the network. Since for the newly added two datasets, intermediate outputs have larger feature size, more information is shared with the server nets. In this case, the reconstructed images will certainly contain more details (in particular the contours used for classification). In fact, you can even see the contours with a randomly initialized head after reconstruction.
> > >
> > > The experiments empirically show that Ensemble network works at least much better than a single network on protecting the details of input. After all, a totally indistinguishable reconstruction is just not possible to be achieved with Gaussian noise+1 layer of convolutional layer on images with large size.
> > >
> > > Thank you very much again for your detailed feedbacks in the review! We really appreciate them!

---

### Official Review · Reviewer_Z7Pt · 2023-11-01

**Soundness:** 2 fair
**Presentation:** 2 fair
**Contribution:** 2 fair
**Rating:** 3
**Confidence:** 2

**Summary:**

The paper proposes an ensembling model architecture for collaborative inference to defend against model inversion attacks. The idea is to employ N server-side networks instead of only one. The client will maintain a secret selector that will only choose a part of the server networks to make final predictions.

**Strengths:**

The paper is easy to read, with a comprehensive introduction of the background & an easy-to-understand presentation of the approaches.

**Weaknesses:**

1. The paper spends too many pages on background introduction. Only until page 6, the authors start to introduce their own approaches...   Also, the paper particularly mentions the privacy of NLP in Section 3.2. But this seems to be irrelevant to the paper... The evaluation is also performed on Cifar-10 with ResNet-18. This can confuse the audience a lot.

2. The evaluation is weak. Only ResNet-18 and Cifar10 are considered. It is also unknown how the effectiveness of the proposed approaches varies with different choices of N and P.

3. Since multiple networks are used in the approaches, the redundancy of computation and the efficiency cost would be a concern. The authors should carefully discuss it.

**Questions:**

I am also confused about the methodology. In my opinion, the idea of multiple networks + selector is basically making the server-side networks wider, and the selector can be simply deemed as an additional layer that projects the outputs from the wider network to lower dimensional representations. From this perspective, this does not fundamentally change the game. Adaptive attackers that know the architecture, can simply model the redundancy and selector as well, and try to reconstruct the additional selector layer.

Can the authors also make some clarifications on whether adaptive attacks may make the proposed defenses less effective?

---

> ### Author Response · Authors · 2023-11-21
>
> Dear reviewer,
>
> We want to thank you for your valuable suggestions, and we try to address your concerns here.
>
> Weakness1: too many pages on the background discussion
>
> Thank you very much for your suggestion. We have moved the NLP part to the Appendix. We still want to include them as NLP is an increasingly important area, and privacy in this area should also be researched.
>
> Weakness2: weak evaluation
>
> We apologize for the weak evaluation presented in the paper. We have provided two additional experiments on datasets with larger intermediate feature size. Regarding the second part of the concern, P does not affect the model’s performance in accuracy. We have experiments with P=3,4,5 for three datasets, and the proposed framework performs as good as the unprotected network. N is associated with the time complexity for inference and reconstruction. As discussed in the response to all reviewers, the proposed framework has a lower-bound inference complexity of N, with hypothesized reconstruction complexity of 2^N.
>
> Weakness3: time complexity
>
> We have now included the discussion of time complexity in the main text.
>
> Question1: why is the proposed model effective?
>
> Please see the response to all reviewers - Major Concern 2a

---

> > ### Comment · Reviewer_Z7Pt · 2023-11-22
> > **Thanks for your reply**
> >
> > Dear Authors,
> >
> > I am not convinced by the proposed methodology. The proposed defense only makes the whole pipeline more complicated and slow in computation. Also, as I suggested before:
> > > The idea of multiple networks + selector is basically making the server-side networks wider, and the selector can be simply deemed as an additional layer that projects the outputs from the wider network to lower dimensional representations
> >
> > It does not appear to make the game fundamentally different. The authors' responses still do not address this fundamental question. Thus, I will keep my initial judgment.
> >
> > Thanks!

---

### Author Response · Authors · 2023-11-21
**Response to major concerns from reviewers**

We want to express our sincere appreciation to all the reviewers and their insightful comments. In this response, we address some of the main concerns that are shared among most of the reviewers, and we will address the individual concerns separately under each review.

Major concern 1: the lack of evaluation to support the proposed framework

All reviewers raised concerns on the lack of evaluations, in particular on images with higher resolution. We acknowledge the lack of evaluations in our experiment. In addition to the existing CIFAR10, we have added experiments with two datasets: CIFAR100 and a subset of CelebA-HQ [1]. Due to the shortness in time, we are not able to conduct experiments on ImageNet nor networks other than ResNet-18, as some of the reviewers have requested. However, with these three datasets, we are able to show how the proposed defense networks react to different feature sizes of the intermediate output from $M_{c,h}$. For CIFAR100, we modified the network by removing the max-pooling layer after the first convolutional layer, making the feature size 32x32. For CelebA-HQ, we follow the data normalization technique used in the paper and reshape the image into 256x256. This gives us an intermediate feature size of 64x64. From the new experiments, we see that the proposed framework is still effective. Quantitatively speaking, Ensembler achieves comparable performance with a randomly initialized $M_{c,h}$, while maintaining the same level of accuracy with a trained, un-protected $M_{c,h}$. Visually speaking, while the protection seems weak for images with higher resolution, it is mainly due to the limit of the simple Gaussian noise, as illustrated with the random $M_{c,h}$ case. Also, due to time concerns, we used standard training setting on these new datasets and did not perform any hyperparameter searching. During the experiments, we have found that the ensemble model could have been regularized more aggressively to increase the protection.

Major concern 2: the lack of an “adaptive attacker” that tries to learn the noise and the selector.

We apologize for any confusions and lack of clarity in the paper. But in the previous version, the N=10 case is the adaptive attacker. We have made the following adjustments to clarify the intuition behind the proposed framework, as well as the experiments on adaptive attacker:

We add a section (Section 4.3) to specifically discuss the intuition behind the proposed framework. We also include the summary here for the convenience of the reviewers. The purpose of the two-staged training algorithm is to ensure that the attacker is not able to learn the selector with its shadow network. Through the first stage of training, we obtain N different models that have distinctive weights, yet all of them are able to make comparative predictions on the dataset. An arbitrary ensemble of P out of the N networks will form a new network, whose $M_{c,h}$ will be distinctive from networks under a different combination. Thus, with N networks in the first stage of the algorithm, we will have $2^N$ different possible $M_{c,h}$ that could be the valid answer to the shadow network. When the attacker tries to train an adaptive attacker, it will fail. Each S’ is a “valid” activation that results in a different $M_{c,h}$, but only one of them is a “correct” S that is used by the client.
This also leads to the discussion of time complexity that some reviewers are interested in. In the previous version, we put the discussion on time complexity in the Appendix, mainly because we are not able to come up with a formal proof for it. In this version, we have moved it with the discussion about intuition of the algorithm. It is not hard to see that under a 1-core GPU, the time complexity of Ensembler is N times compared to no protection, with marginally more computations for the client. There is almost no extra data transportation. However, under modern server settings with multiple GPUs or even with multiple clusters, since the framework is very friendly to parallel execution, the extra server computation cost is negligible, especially considering the exponential reconstruction complexity.
We have renamed the N=10 to adaptive attacker and make a comment in the experiment setup for clarity.

[1] Na, D., Ji, S., Kim, J. (2022). "Unrestricted Black-Box Adversarial Attack Using GAN with Limited Queries." In Proceedings of the European Conference on Computer Vision (ECCV), 2022.

---

### Meta-Review · Area_Chair_ux53 · 2023-12-12

**Metareview:**

This paper develops a new framework for enhancing defenses against model inversion attacks in collaborative inference scenarios. Specifically, it employs multiple server-side networks and a secret selector on the client side to ensure that only a subset of server networks contribute to the final prediction. Experiments on CIFAR-10 with ResNet-18 are provided to support its effectiveness. While reviewers find this paper interesting to read, they raise several major concerns: 1) The empirical evaluation is deemed insufficient, limited to CIFAR-10 with ResNet-18, and lacking in diverse scenarios or datasets; 2) The setup of threat models is questionable and not compelling; 3) no discussions about adaptive attacks; and 4) more computations are introduced. The rebuttal is considered, but reviewers are not fully convinced and unanimously recommend rejecting this paper.

The authors are encouraged to carefully address these concerns and make a stronger submission next time.

**Justification For Why Not Higher Score:**

The reviewers have significant concerns about this paper and do not believe it is ready for publication at this time.

**Justification For Why Not Lower Score:**

N/A

---

### Decision · Program_Chairs · 2024-01-16

Reject